# *Cutibacterium avidum*: A Potent and Underestimated Pathogen in Prosthetic Hip Joint Infections

**DOI:** 10.3390/microorganisms12030432

**Published:** 2024-02-20

**Authors:** Johanna Karlsson, Nina Kamenska, Erika Matuschek, Holger Brüggemann, Bo Söderquist

**Affiliations:** 1NU Hospital Group, Department of Infectious Diseases, 461 73 Trollhättan, Sweden; 2Department of Infectious Diseases, Institute of Biomedicine, University of Gothenburg, 405 30 Gothenburg, Sweden; 3NU Hospital Group, Department of Clinical Microbiology, 461 73 Trollhättan, Sweden; nina.kamenska@vgregion.se; 4EUCAST Development Laboratory, Clinical Microbiology, Central Hospital, 352 34 Växjö, Sweden; erika.matuschek@kronoberg.se; 5Department of Biomedicine, Aarhus University, 8000 Aarhus, Denmark; brueggemann@biomed.au.dk; 6Faculty of Medicine and Health, School of Medical Sciences, Örebro University, 701 82 Örebro, Sweden; bo.soderquist@oru.se

**Keywords:** *Cutibacterium avidum*, prosthetic joint infections, whole-genome sequencing, molecular epidemiology

## Abstract

*Cutibacterium avidum* has recently been reported as a rare cause of prosthetic joint infections (PJIs), contrary to *Cutibacterium acnes*, which is well established as a cause of PJIs, especially in shoulder arthroplasties. Two specific risk factors for PJI due to *C. avidum* have been reported: obesity and the skin incision approach. Here, we report four cases of hip PJIs caused by *C. avidum* admitted over a 30-month period at a single center. Whole-genome sequencing revealed that the four *C. avidum* strains were all individual strains and did not originate from a common source, such as an outbreak. Antibiotic susceptibility tests showed that the isolates were fully susceptible, and none carried known antibiotic resistance genes. In conclusion, the occurrence of four cases of PJI caused by *C. avidum* over a limited time at a single center may indicate that this pathogen is underestimated and is either emerging or more common than previously recognized. The patients presented overt signs of infection during surgery, indicating that *C. avidum* is a virulent pathogen. None of the previously reported risk factors for *C. avidum* PJI applied to these patients as only one was obese and none were operated on using a direct anterior skin incision approach.

## 1. Introduction

Total hip replacement is one of the most important medical technologies established in recent decades. It has significantly relieved pain and improved or restored the quality of life for millions of patients. Although arthroplasty is considered a safe procedure, there are sometimes complications, the most feared of which is prosthetic joint infection (PJI). Although rare, PJIs result in significant suffering for patients, increased morbidity and mortality [1], and high costs for healthcare providers.

The most common etiological agents of PJIs are staphylococci, both *Staphylococcus aureus* and coagulase-negative staphylococci (CoNS), predominantly *Staphylococcus epidermidis*, which accounts for 55–75% of cases [2,3]. Other pathogens associated with PJIs include Enterobacterales, for example, *Escherichia coli*, alpha- and beta-hemolytic streptococci, enterococci, and anaerobes, especially *Cutibacterium acnes* [2].

*Cutibacterium acnes* (formerly known as *Propionibacterium acnes*) is a well-established cause of PJIs [4,5,6,7], especially after shoulder arthroplasties [8]. Other *Cutibacterium* spp., such as *Cutibacterium granulosum*, have also occasionally been described as a cause of PJIs [9]. In addition, *Cutibacterium avidum* has recently been reported as a cause of PJIs [7,10,11,12,13,14]. Nevertheless, data regarding *Cutibacterium avidum* are rather scarce. Analysis of a *Cutibacterium avidum* genome has revealed the presence of an exopolysaccharide (EPS) biosynthesis gene cluster, and electron and atomic force microscopy studies have confirmed the presence of an EPS-like structure surrounding cells of *C. avidum* [15].

Together with CoNS and *Corynebacterium* spp., *Cutibacterium* spp. are the main constituents of the microbiota of the skin. *C. avidum* is often isolated from moist areas of the skin, such as the nares, axilla, rectum, and predominantly the groin [16].

The first two cases of hip PJIs due to *C. avidum* were reported in 2016 by Wildeman et al. [10]. In both cases, the surgical procedure used the anterior approach and both patients were obese (BMI ≥ 30 kg/m^2^) with body mass indexes (BMIs) of 37 and 38, respectively. We hypothesized that this anterior approach in close proximity to the groin could constitute a risk factor for deep-seated infections caused by *C. avidum.* This hypothesis was supported by a study by Böni et al. [17], who noted increased recovery of *C. avidum* from the moist skin folds in the groin of obese patients. The characteristics of *C. avidum* PJIs were further reported in two studies in 2018 describing 15 [11] and 13 patients [12], respectively. All but two of these patients underwent hip arthroplasty.

In this paper, we report four cases of hip PJIs caused by *C. avidum.* The patients were admitted over a 30-month period at a single center. All four patients presented serious to fulminant infections. Genomic data for these geographically and temporally related *C. avidum* isolates are provided.

## 2. Case Presentations

### 2.1. Case 1

A 55-year-old man with a history of smoking, atrial flutter, and obesity, BMI 35, underwent uncemented total hip arthroplasty of the left hip due to severe coxarthrosis in April 2020. The current medication was bisoprolol and candesartan. The surgery was performed with an anterolateral approach with a good functional outcome. However, the patient had residual pain from the hip, and acute-phase parameters were slightly elevated four months after the surgery (CRP 18 mg/L, ESR 24 mm/h). No radiological signs of prosthesis dysfunction were found, and a puncture of the hip yielded a dry tap.

However, due to increasing pain and a gradual rise in CRP levels (maximum 32 mg/L), a deep periprosthetic joint infection was suspected. An X-ray 8 months after the primary operation revealed radiolucent zones surrounding the stem and the cup, and joint fluid samples taken by arthrocentesis were culture-positive for C. avidum. Three months later, a one-stage exchange procedure using an uncemented Corail device was performed. During surgery, an obvious infection was observed with pus under high pressure in the joint cavity and necrotized soft tissue. Five tissue biopsies were collected from the acetabulum and femur, and the patient started antibiotic treatment with 3 g of benzylpenicillin and 1 g of vancomycin t.i.d. Collatamp with a total of 80 mg of gentamicin was applied to the wound. Tissue samples showed monomicrobial growth of *C. avidum* in all ten biopsies. The antimicrobial susceptibility test (AST) pattern is shown in Table 1.

Vancomycin was discontinued after seven days when the results of the tissue cultures were reported. Penicillin G was administered for another seven days with the addition of 450 mg of clindamycin orally t.i.d. The patient was discharged from the hospital after two weeks with an oral combination treatment of clindamycin as above and 1 g of amoxicillin t.i.d. At a follow-up visit six weeks after the revision, CRP was normalized at 2.6 mg/L, and there were no clinical signs of infection. Antibiotic treatment was discontinued after a total of 4.5 months. Further follow-up for almost two years has been uneventful.

### 2.2. Case 2

A previously healthy and physically active 57-year-old man without any medication and with normal body weight, BMI 25, received an uncemented prosthesis in his right hip at a private orthopedic clinic in June 2020 due to severe coxarthrosis. The surgery was performed with a posterior skin incision, and the procedure was considered uncomplicated. At follow-up after 6 weeks, the wound was healed, and the patient was experiencing significantly less pain.

Approximately four months after the operation, the patient had an onset of migrating body pain, recurring low-grade fever, epigastralgy, and increasing pain in the right hip. An extensive medical investigation was performed by his general practitioner. Laboratory tests were normal except for slightly elevated acute-phase parameters with a CRP of 19 mg/L and ESR of 33 mm/h. The patient was admitted for gastroscopy, through which eosinophilic esophagitis was diagnosed. Treatment with oral steroids was given for three months with a proper effect on the esophagitis but no improvement of general inflammatory symptoms or hip pain. Arthrocentesis of the right hip was performed in January 2021. The joint fluid showed an increased level of leucocytes (85 × 10^9^/L), and the bacterial culture was positive for *C. avidum.* A one-stage exchange revision using an uncemented stem and cup (Corail, Continuum) was performed in April 2021, ten months after the primary operation. Intraoperatively, clear signs of infection were observed with pus present in the joint. The patient had not received any antibiotics preoperatively but was started on 3 g of penicillin G and 1 g of vancomycin, both t.i.d. intraoperatively. A Septocoll fleece containing a total of 140 mg of gentamicin was also applied to the wound.

Thirteen of the fifteen tissue cultures obtained intraoperatively showed growth of *C. avidum*. *Staphylococcus saccharolyticus* was detected in one sample, and one was culture-negative. The AST pattern of *C. avidum* is shown in Table 1.

Intravenous antibiotic treatment was discontinued on day 8 postoperatively and replaced by 1 g of oral amoxicillin t.i.d. At this stage, there was no remaining wound leakage, and the patient was discharged from the hospital. At a follow-up visit three months postoperatively, the wound had healed, and the patient was experiencing no pain from the hip. CRP was normalized at 1.8 mg/L, and the antibiotic treatment was stopped. Follow-up for two years has been uneventful.

### 2.3. Case 3

The third patient was a 64-year-old man, slightly overweight with a BMI of 27 and a history of acute lymphatic leukemia in 2013 but in complete remission after chemotherapy. This was followed by secondary hypogammaglobulinemia, which was treated with intravenous immunoglobulin monthly until June 2021, when it was stopped due to an allergic reaction. He had septic arthritis of his right ankle in 2019 due to group C streptococci and was treated with amoxicillin for 6 weeks. In 2019, the patient was diagnosed with prostate cancer and received curative radiotherapy and hormonal treatment with bicalutamide. The radiation was complicated by proctitis and gut bleeding. He also had a history of deep venous thrombosis in his right leg. In 2020, a small kidney tumor was found on a CT scan, requiring regular radiological follow-up. Furthermore, the patient had been diagnosed with spinal stenosis, hypertension, and depressive symptoms.

In November 2021, he was admitted to the Department of Orthopedics due to long-lasting pain in his right hip. An MRI revealed arthrosis and caput necrosis, and the patient underwent total hip arthroplasty. The medication administered was amlodipine, venlafaxine, mirtazapine, oxycodone, and ibuprofen. He received an uncemented Lubinus prosthesis in February 2022. The surgery was performed with an anterolateral approach and was complicated by large bleeding of 1000 mL. Osteonecrosis of the caput femoris was observed as well as massive synovial inflammation.

The patient recovered well and was discharged from the hospital after three days. However, on day 10 postoperatively, he was readmitted due to the sudden onset of severe pain in the right groin and fever. Ultrasound examination revealed an 11 × 10 × 6 cm abscess. Puncture of the hip yielded pus, which was later found to be culture-positive for *C. avidum*. CRP was 172 mg/L. Antibiotic treatment was started with piperacillin/tazobactam, but after spontaneous wound rupture with drainage of the abscess, he became afebrile, and antibiotics were discontinued after one day in order to enhance the possibility of adequate microbiological findings in tissue cultures from the following revision surgery.

The patient was reoperated on day 17 after the primary hip surgery and seven days after the onset of infectious symptoms, with debridement and implant retention (DAIR) and exchange of the modular caput. The wound and the joint cavity revealed an aggressive infectious picture with fibrin and nonviable tissue. Ten tissue samples were collected from the interfaces between the prosthesis stem and femur and caput and acetabulum, and the patient restarted antibiotic treatment with 4 g/0.5 g of piperacillin/tazobactam t.i.d. The surgical wound was thoroughly debrided and packed with Collatamp with a total of 500 mg of gentamicin.

The patient was afebrile after five days with a slowly decreasing CRP. All tissue cultures showed monomicrobial growth of *C. avidum.* The AST results are shown in Table 1. After 14 days of intravenous treatment, antibiotics were shifted to 1 g of oral amoxicillin and 450 mg of clindamycin, both t.i.d. The patient was discharged from the hospital with a planned total antibiotic treatment of three months. At follow-up after 14 days, there was still wound secretion and skin redness; however, they were decreasing, and the antibiotic treatment was unchanged. Immunoglobulin levels were measured and found to be very low, with a total IgG of 0.94 g/L, and the patient was restarted on immunoglobulin replacement therapy, now administered subcutaneously once a week without complications.

After three months, the wound had healed, but acute-phase parameters were still slightly elevated. Because of this elevation and the patient’s immunosuppression, oral antibiotic treatment was prolonged for a total duration of six months postoperatively. The follow-up thereafter was uneventful with low CRP values (below 10 mg/L except when controlled during episodes of respiratory tract infections related to the patient’s immunodeficiency) and no patient-reported pain.

### 2.4. Case 4

The patient was a 62-year-old man with a history of Parkinson’s disease, asthma, hypertension, and hypercholesterolemia. He was overweight with a BMI of 29.5, and his ongoing medication was levodopa/carbidopa/entacapone, pramipexole, candesartan/hydrochlortiazide, atorvastatin, desloratadine, budesonide and salbutamol inhalations, naproxen, and paracetamol.

The patient underwent total hip arthroplasty due to severe coxarthrosis in December 2022 and received an uncemented prosthesis with a Bimetric stem. The surgery was performed with an anterolateral approach and was uncomplicated. He was discharged from the hospital two days later.

On postoperative day five, the patient was readmitted due to fever and increasing pain from the operated hip for three days. CRP was 230 mg/L. There was redness and secretion from the wound. An ultrasound revealed an underlying abscess, 60 × 26 × 12 mm, and a puncture was performed that yielded sanguinolent fluid. The fluid culture showed growth of *Staphylococcus capitis.* The patient was continuously febrile and shivering, but antibiotic treatment was postponed since he was not septic. Blood cultures were negative, but *Klebsiella pneumoniae* was found in the urine.

DAIR was performed four days after admittance to the hospital. The intraoperative tissue showed a massive infective picture with leakage of seropurulent fluid, necrotized fascia, and loss of substance. The liner was removed, and extensive soft tissue debridement and thorough irrigation were performed. The stem was found to be well fixed. Local gentamicin (240 mg) and vancomycin (1 g) were applied in STIMULAN Rapid Cure (Biocomposites Ltd., Keele, UK). A total of eleven tissue biopsies were collected, which all showed monomicrobial growth of *C. avidum*. The AST pattern is shown in Table 1. The patient intraoperatively started empirical antibiotic treatment with 4 g/0.5 g of piperacillin/tazobactam and 1 g of vancomycin t.i.d. Vancomycin was discontinued after six days, when the result of the tissue cultures was available, and piperacillin/tazobactam was continued up to ten days. At this point, the patient was afebrile, and CRP decreased to 15 mg/L. There was still some wound leakage, but it slowly diminished. Antibiotics were shifted to 1 g of amoxicillin and 450 mg of clindamycin perorally, both t.i.d. The patient was discharged from the hospital three days later.

At follow-up after four weeks, the wound was still leaking, and a small granuloma had developed in the proximal part of the scar. Antibiotics were continued unchanged, and after three months, the acute-phase parameters were normalized. The patient experienced no pain and had successfully started rehabilitation. However, the granuloma was still present with some secretion, and the decision was made to prolong antibiotic treatment to six months. By then, the secretion had stopped, the granuloma had almost disappeared, and antibiotics were discontinued. Inflammatory parameters remained normal; however, the granuloma was still present, and three months after antibiotic discontinuation, a discrete wound secretion was again seen. Since the patient had no pain and there were no radiological or laboratory signs of infection or prosthesis dysfunction, he is at present attending frequent follow-ups.

## 3. Methods

### 3.1. Bacterial Isolates

Intraoperative tissue samples were cultured at the clinical microbiological laboratory on Columbia blood agar base (OXOID, Basingstoke, UK) supplemented with 5% horse blood (Håtunalab, Bro, Sweden) at 36 °C anaerobically for up to 5 days, on medium GC agar base (OXOID, Basingstoke, UK) supplemented with 1% soluble hemoglobin powder (OXOID, Lowell, MA, USA), Vitox supplement powder (OXOID, UK), and Vitox hydration fluid (OXOID, Basingstoke, UK) at 36 °C in 5% CO_2_ for up to 5 days, and in enrichment broth (Fastidious Anaerobe Broth; FAB) (Neogen, Bridgend, UK) at 36 °C for up to 10 days. Species identification was performed by matrix-assisted laser desorption/ionization time-of-flight mass spectrometry (MALDI-TOF MS) (Vitek MS and Myla V4.8; BioMérieux SA, Marcy L’Etoile, France).

### 3.2. Antibiotic Susceptibility Testing

Minimum inhibitory concentrations (MICs) were determined by a gradient test (Etest, bioMérieux, Marcy l’Etoile, France) according to the manufacturer’s instructions. Nine antibiotics were tested: benzylpenicillin, piperacillin/tazobactam, meropenem, moxifloxacin, vancomycin, erythromycin, clindamycin, rifampicin, and metronidazole. Antibiotic susceptibility testing was performed on Brucella blood agar with hemin and vitamin K with 1.0 McFarland suspensions of bacteria in NaCl and incubation at 36 °C under anaerobic conditions for 24 h. Clindamycin MICs were confirmed after 48 h of incubation.

### 3.3. Genome Sequencing and Analysis

The MasterPure™ Gram Positive DNA Purification Kit (Lucigen, Middleton, WI, USA) was used according to the manufacturer’s instructions for genomic DNA extraction. DNA quality and yield were checked by agarose gel electrophoresis along with concentration determination using the Qubit^®^ dsDNA HS Assay Kit (Life Technologies GmbH, Darmstadt, Germany). Illumina shotgun libraries were prepared using the Nextera XT DNA Sample Preparation Kit and subsequently sequenced on a MiSeq system using the v3 reagent kit with 600 cycles (Illumina, San Diego, CA, USA), as recommended by the manufacturer. Quality filtering was performed with version 0.39 of Trimmomatic [18]. Assembly was performed with version 3.15.2 of the SPAdes genome assembler software [19]. Version 2.2.1 of Qualimap was used to validate the assembly and determine the sequence coverage [20]. The four *C. avidum* strains were sequenced with a genome coverage of 212 to 333. Default parameters were used for all mentioned software unless otherwise specified. The draft genome sequences were deposited in GenBank with accession numbers JAXCLG000000000, JAXCLH000000000, JAXCLI000000000, and JAXCLJ000000000.

Gene prediction and annotation of all genomes were performed with RAST [21]. For phylogenomic analyses, the core genome was identified and aligned with the Parsnp program from the Harvest software package (v1.7.4) [22]. *C. avidum* genomes available from GenBank (status from August 2023) were used along with the four *C. avidum* genomes from this study to build a core-genome-based phylogeny. Reliable core-genome single-nucleotide variants identified by Parsnp were used to reconstruct the genome-based phylogeny using FastTree 2 [23]. Phylogenetic trees were visualized (as unrooted trees) using the Interactive Tree of Life [24]. BRIG was used for genome comparison and visualization [25].

## 4. Results

### 4.1. Genome Sequencing of C. avidum Isolates

The genome size of the isolates ranged from 2502 to 2591 kbp. Annotation-predicted coding sequences (CDS) ranged from 2385 to 2468 CDS (Table 2).

A core-genome comparison was carried out. Two main phylogenetic lineages within the *C. avidum* population can be detected (Figure 1). All sequenced PJI isolates belonged to the same lineage. This lineage also contained strains isolated from other PJI cases, including isolates T13, T14, and T15 [10] and FMS2275 and FMS4815 [26]. There were also isolates associated with other diseases such as TM16, isolated from radical prostatectomy specimens [15]. Single-nucleotide variants (SNVs) ranging from 17,167 to 42,685 were detected among these isolates.

Strain individualities were illustrated by direct genome comparisons, revealing isolate-specific genomic regions and regions with sequence similarity below 99% (Figure 2).

### 4.2. Antibiotic Susceptibility Testing

All four *C. avidum* isolates obtained from the four patients displayed no resistance to tested antibiotics, except metronidazole (Table 1).

## 5. Discussion

Only a few cases of PJI due to *C. avidum* have been reported. The first two cases were reported by us in 2016 [10]. In 2018, two case series [11,12] of 12 and 15 patients, respectively, were published. These patients were collected over a >10-year study period. However, PJIs due to *C. avidum* are probably more common than previously recognized. In the study by Simon et al. [27], 10/178 (5.1%) of the cultures from PJI patients yielded *C. avidum*. In the present study, a cluster of four cases with a confirmed diagnosis of PJI due to *C. avidum* from April 2020 to December 2022 was identified. These cases originated from one administrative region in Western Sweden with approximately 300,000 inhabitants.

In previous reports, at least two specific risk factors were identified: the BMI of the patient and the surgical approach. The first two cases of *C. avidum* reported in the previous study had a BMI of 37 and 38, respectively [10]. The 12 patients reported by Achermann et al. [12] had a median BMI of 34 with a range from 27.9 to 40.6, and 9 of 12 were obese. Zeller et al. [11] similarly found the median BMI to be 35 (range 24–40), and 11 of 15 (73%) had a BMI ≥ 30. However, of the four patients in the present case report, only one was obese (BMI 35), and the median BMI was 28, with a range of 25 to 35. This indicates that obesity might not be a strong risk factor.

The skin incision approach may also constitute a risk factor. A direct anterior approach, which may be more anatomical and thus less invasive than an anterolateral, lateral, or posterior approach, is widely used. However, an anterior skin incision is adjacent to the groin, which is a highly colonized site of the body [28]. *C. avidum* is often isolated from moist areas of the skin, such as the groin [16], in contrast to *C. acnes*, which mostly occurs at sebaceous sites (face, back, and thoracic) that are dry but lipid-rich. Thus, an incision close to the groin area may result in an increased risk of contamination of the surgical wound with skin commensals such as *C. avidum* and thereby also an increased risk for PJIs [29]. This issue was investigated in a study in which the direct anterior and lateral transgluteal approaches were compared regarding the risk of infection and the microbial spectrum of PJI [30]. In that nonrandomized study, no increased risk of PJI was found when applying the direct anterior approach. However, microorganisms representing anaerobic skin microbiota were found only among PJI patients who had undergone hip arthroplasty by an anterior approach. Furthermore, an increased rate of skin colonization of *C. avidum* in the groin compared with the anterior or lateral thigh has been demonstrated in patients scheduled for primary hip arthroplasty [17]. In none of the four patients in the present case report was an anterior skin incision used. In three cases, an anterolateral approach was used, and in one case, a posterior approach was used. This suggests that a direct anterior skin incision approach is not a particular risk factor for *C. avidum* PJI.

*C. avidum* is a skin commensal and may represent a low-virulence pathogen presenting as delayed or late chronic foreign-body infections [14]. However, the clinical presentation of the four cases presented here was rather acute and fulminant with overt signs of infection intraoperatively. Two patients were septic, which raised suspicion of an infection with highly virulent bacteria such as *Staphylococcus aureus* or ß-hemolytic streptococci. One of these patients was immunosuppressed with low immunoglobulin levels after a history of acute lymphatic leukemia, although in complete remission for several years. However, the other patient was immunocompetent, as far as is known.

All four patients received cloxacillin as prophylaxis since it is the drug of choice as prophylaxis for arthroplasty surgery in Sweden and is used in 93.4% of primary arthroplasties. Only if a history of allergy is present will clindamycin be used (4.3%) [31]. Cephalosporins and other antibiotics were used in the remaining cases. However, this prophylactic strategy may not be optimal for the prevention of PJI caused by *Cutibacterium* spp. since, at least in vitro, the MIC values are 3–4-fold higher for cloxacillin compared to benzylpenicillin [32]. Since *C. acnes* is an important etiological agent in shoulder PJIs, benzylpenicillin has increasingly been added to routine prophylaxis with cloxacillin for shoulder prosthetic surgery in Sweden. A register-based study [33] using data from the Swedish Shoulder Arthroplasty Register found that cloxacillin prophylaxis was associated with a significantly increased relative risk of reoperation for infection compared to the combination of cloxacillin and benzylpenicillin. In addition, since cloxacillin has a short half-life, is highly protein-bound, and the free concentration could be affected by the patient’s BMI, serum albumin, and renal function, it may constitute a risk of subtherapeutic concentrations of cloxacillin during surgery in specific patients. In hip arthroplasties employing an anterior skin incision approach, especially in obese patients, the addition of benzylpenicillin should be considered. Additionally, cephalosporin is an option, although the MIC for *C. acnes* is higher than that of benzylpenicillin [32]. Local application of antibiotics, preferentially gentamicin in sponges, as commonly used in our patients, is not a beneficial treatment option for anaerobes, such as *Cutibacterium* spp., since the aminoglycosides inherently lack activity against anaerobic bacteria. A better choice in these cases is a combination of aminoglycosides and vancomycin (in, for example, calcium sulfate ± hydroxyapatite).

The four *C. avidum* strains analyzed here were all individual strains. They did not originate from a common source, such as an outbreak scenario in a hospital setting. Most likely, these four strains originated from the skin of the individual patients. However, even though they were individual strains, they all belong to the same *C. avidum* phylogenetic clade. This clade includes several other PJI isolates, which implies that strains from this clade are equipped with genes/functions that are advantageous for their opportunistic pathogenic phenotype. Such functions could be adhesion/biofilm properties, host immune evasion mechanisms, intra-/interspecies competition, and other fitness functions, such as nutritional and metabolic peculiarities, allowing in-host survival and replication.

None of the *C. avidum* isolates were resistant to any of the tested antibiotics according to the EUCAST’s guidelines (www.eucast.org/fileadmin/src/media/PDFs/EUCAST_files/Guidance_documents/When_there_are_no_breakpoints_20230630_Final.pdf accessed on 9 February 2024). None of the isolates carried known antibiotic resistance genes (ARGs). In previous studies, *C. avidum* strains were found to carry ARGs related to macrolide–clindamycin resistance; in particular, the ARG erm(X) (located on Tn5432) was found [34,35,36]. A recent study also found in vivo emergence of resistance to a fluoroquinolone (levofloxacin) and rifampicin [26]; genomic mutations in *gyrA* and *rpoB* could be identified that were responsible. Such mutations were absent in the four strains sequenced here.

## 6. Conclusions

We report four cases of PJI caused by *C. avidum* during a limited time at a single center, which may indicate that this pathogen is either emerging or is underestimated and more common than previously recognized. The patients presented with fulminant infections, including overt signs of infection intraoperatively, suggesting virulent characteristics of *C. avidum*. These patients did not have previously reported specific risk factors as only one was obese, and none had undergone surgery using an anterior skin incision approach.

## Figures and Tables

**Figure 1 microorganisms-12-00432-f001:**
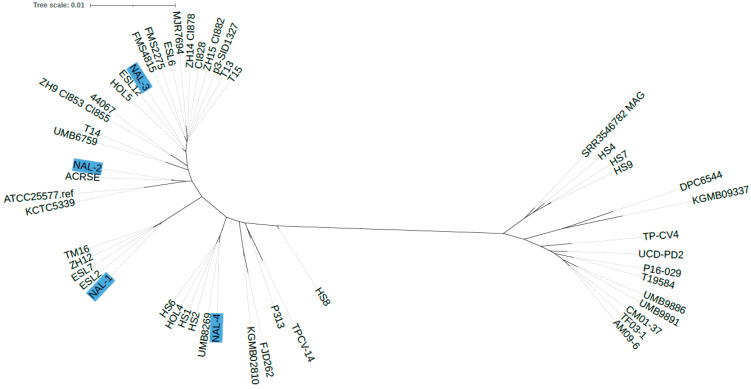
Phylogenetic comparison of the core genome of *C. avidum*. Core-genome SNP analysis and phylogenetic reconstruction were performed with Parsnp. The isolates sequenced here are highlighted in blue. The other genomes were taken from GenBank (NCBI). In total, 52 genomes were used for core-genome-based phylogenetic reconstruction.

**Figure 2 microorganisms-12-00432-f002:**
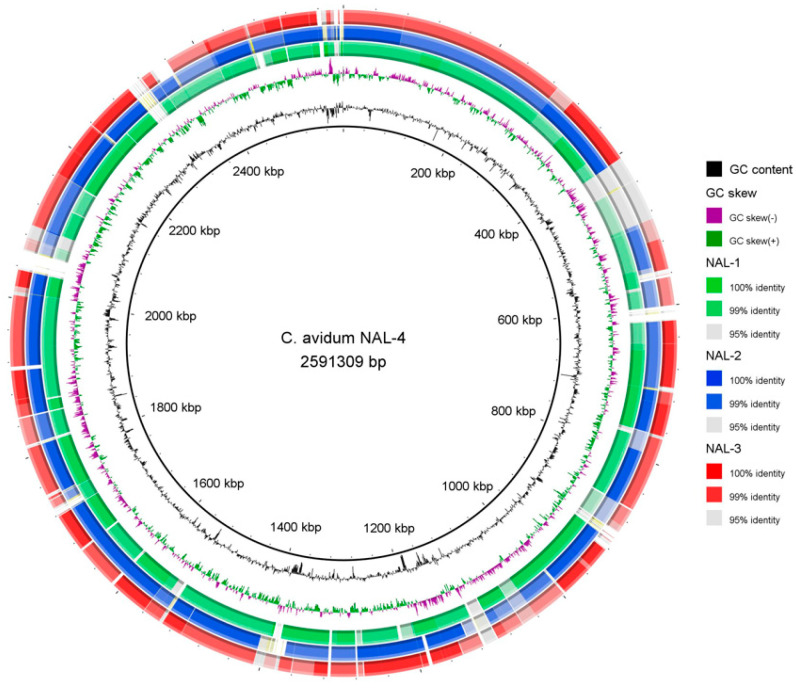
Genome comparison of *C. avidum* NAL-4 with *C. avidum* NAL-1 to NAL-3. The genome of *C. avidum* NAL-4 was compared with the other three sequenced genomes by BLASTN. The analysis highlights the genomic differences between the isolates, with NAL-4-specific genomic regions (three larger regions) and regions with sequence similarity below 99%. The image was created with BRIG.

**Table 1 microorganisms-12-00432-t001:** MIC values (mg/L) for nine antibiotics for *Cutibacterium avidum* isolated from four patients with prosthetic hip joint infections.

Antibiotics	Case 1	Case 2	Case 3	Case 4
Penicillin G	0.125	0.064	0.064	0.125
Piperacillin/Tazobactam	1.0	1.0	1.0	2
Meropenem	0.125	0.125	0.064	0.125
Moxifloxacin	0.125	0.125	0.125	0.125
Vancomycin	1	1	0.5	1
Erytromycin	0.032	0.032	0.032	0.016
Clindamycin	0.032	0.032	0.032	0.032
Rifampicin	0.008	0.004	0.004	0.004
Metronidazole	>256	>256	>256	>256

**Table 2 microorganisms-12-00432-t002:** *Cutibacterium avidum* genomes sequenced in this study.

Strain	Coverage	Contigs	GC	CDS	Length (bp)	GenBank Accession
NAL-1	212	18	63.4	2405	2,506,293	JAXCLG000000000
NAL-2	333	17	63.4	2385	2,510,992	JAXCLH000000000
NAL-3	323	16	63.4	2413	2,503,367	JAXCLI000000000
NAL-4	290	32	63.4	2468	2,591,309	JAXCLJ000000000

## Data Availability

All the data that support the findings of the case report have been provided in the article and are also available from the corresponding author upon reasonable request.

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
