# Peer review of "Cutibacterium avidum: A Potent and Underestimated Pathogen in Prosthetic Hip Joint Infections"

_microorganisms, 2024, doi:10.3390/microorganisms12030432_

Round 1
Reviewer 1 Report
Comments and Suggestions for Authors
Dear authors:
Thank you for your interesting article.
The third patient had been shown to have pus in his prosthetic hip joint (lines 151-152) and suffered from a large abscess. As per the text, antibiotic treatment has been discontinued after wound rupture with drainage of the abscess; antibiotics have been discontinued after one day of treatment. Only 17 days later, when a DAIR procedure has been performed, antibiotics have been started again, based on "anaggressive infectious picture" intraoperatively. What else could have been expected, given both the putrid hip joint puncture and the abscess? Why has the antimicrobial treatment been interrupted for 17 days?
Lines 232-233: "Minimum inhibitory concentrations (MICs) where determined by a gradient test (Etest, bioMerieux, Marcy l'Etoile, France) according to EUCAST guidelines (www.eucast.org)."
To my knowledge, EUCAST has not described susceptibility testing by gradient diffusion for strictly anaerobic bacteria. My suggestion for the lines cited would be: "Minimum inhibitory concentrations (MICs) where determined by a gradient test (Etest, bioMerieux, Marcy l'Etoile, France) and interpreted according to EUCAST guidelines (www.eucast.org)."
In 2020, EUCAST rules (breakpoint table V. 10.0) included MIC breakpoints for "Gram-positive anaerobes"; however, the EUCAST table cited included a dash for ciprofloxacin, levofloxacin, rifampicin, and trimethoprim-sulfamethoxazole. By EUCAST rules, a dash means the agent is not therapeutically beneficial. In 2021, with EUCAST breakpoint table V. 11.0, matters did not change regarding the susceptibility categorization of Gram-positive anaerobes.
In 2022, EUCAST published V. 11.0 of the breakpoint tables. Now, breakpoints were defined for specific anaerobic species or groups, only. The tables include data for Cutibacterium acnes only, and gives MIC breakpoints for benzylpenicillin, piperacillin-tazobactam, meropenem, vancomycin, and clindamycin. If those values were to be applied to the Cutibacterium avidum isolates, the reasoning behind this should be described. Alternatively, the susceptibility categorization might be based on PD-PD breakpoints, which are given in V. 11.0 of the EUCAST rules for benzylpenicillin, piperacillin-tazobactam, meropenem, ciprofloxacin, and levofloxacin, but are lacking for vancomycin, rifampicin, and trimethoprim-sulfamethoxazole.
Please describe which EUCAST breakpoints your have based your susceptibility judgment on in some detail.
EUCAST accepts MIC values arrived at with methods other than the reference methods described in their regulations given the equivalence of the results of the method used in the EUCAST reference method results have been shown.
Given that the medium is a critical factor insusceptibility testing, you should describe why you chose to do susceptibility testing on FAA agar, the medium described by EUCAST for their agar diffusion method, for gradient diffusion with strips from Biomerieux, given that in the Etest package insert, for anaerobes, Brucella agar + 5% blood + vitamin K and hemin (BBA) as defined by CLSI is listed as the medium to be used.
Another deviation from the producer's method description detailed in the article is the use of 0,5 McFarland suspensions of the bacterial strains "in NaCl" (probably a solution of 0.9% of NaCl in sterile water has been used), while the package insert requests the use of iinocula with bacterial densities of 1.0 McFarland, prepared in Brucella, Mueller-Hinton, or Schaedler broth.
Again, please describe the reasons for your deviation from the producer's instructions and the validation of your susceptibility testing method: Reducing the inoculum density must be expected to have a strong influence on the test results.
In lines 293 to 294, you state: "All four C. avidum isolates obtained from the four patients displayed no resistance to the tested antibiotics, except metronidazole (Table 1)." Please explain this statement in light of the lack of breakpoints for ciprofloxacin, levofloxacin, rifampicin, and trimethoprim-sulfamethoxazole.
In addition, based on the current EUCAST breakpoints for C. acnes and the MIC values given in Table 1, two of the strains would be categorized as R to benzylpenicillin and all strains as R to piperacillin-tazobactam, and one strain R to meropenem. Again, please describe on which breakpoints you have based your conclusion on.
Reviewer 2 Report
Comments and Suggestions for Authors
This is a valuable work of research. However, in order to improve its value, I have some suggestions:
- the article seems to lack figures or tables in the provided sections. Including visual data representations; adding these could improve the research visibility over time;
- I suggest the authors to include a more detailed comparative analysis, possibly using phylogenetic trees or genomic comparison data, to strengthen the argument about the pathogen virulence and its underestimation in medical literature;
- a more in-depth analysis of the resistance patterns and their clinical implications would be beneficial
- in a section desired and considered by the authors, please consider citing the current trends in bacterial cultures as described in https://www.mdpi.com/2077-0383/11/8/2238
- a more in-depth description of this pathogens clinical implications would be of importance for the journal
Round 2
Reviewer 2 Report
Comments and Suggestions for Authors
Authors answered all the concerns. The paper is improved.